# Generation of Chimeric African Swine Fever Viruses Through *In Vitro* and *In Vivo* Intergenotypic Gene Complementation

**DOI:** 10.3390/vaccines13050462

**Published:** 2025-04-25

**Authors:** Tomoya Kitamura, Kentaro Masujin, Mitsutaka Ikezawa, Aruna Ambagala, Takehiro Kokuho

**Affiliations:** 1National Institute of Animal Health, National Agriculture and Food Research Organization, Tokyo 187-0022, Japan; kitamurat464@affrc.go.jp (T.K.); masujin@affrc.go.jp (K.M.); mikezawa@affrc.go.jp (M.I.); 2Canadian Food Inspection Agency, National Centre for Foreign Animal Disease, Winnipeg, MB R3E 3M4, Canada; aruna.ambagala@inspection.gc.ca

**Keywords:** African swine fever virus, vaccines, chimerization, *in vitro*, *in vivo*

## Abstract

**Background/Objectives**: African swine fever (ASF), a fatal febrile hemorrhagic disease in domestic pigs and Eurasian wild boars, is caused by ASF virus (ASFV). ASF continues to spread across the globe, causing a significant impact on the world’s pig industry. Recently, highly virulent chimeric ASFV (chASFV) strains with recombined genomes of the p72 genotype I and II viruses have been reported in China, Vietnam and Russia. **Methods**: In order to understand the propensity of ASFV genome for recombination, we attempted to experimentally generate chASFVs both in vitro and *in vivo* employing two distinct attenuated ASFV strains: OUR T88/3 (genotype I) and AQSΔB119L (genotype II). **Results**: When IPKM cells were co-infected with ASFV OUR T88/3 and AQSΔB119L strains, three genetically distinct chASFV emerged. When pigs were inoculated with the individual chASFV isolates, all pigs developed acute ASF. When four pigs were co-infected with ASFV OUR T88/3 and AQSΔB119L, all of them developed acute ASF and died or were euthanized. Three chASFV strains were successfully isolated from splenic homogenates from each pig. **Conclusions**: Our research indicates that genotype I and II chASFV with diverse genomes can be easily generated experimentally both in vitro and *in vivo*.

## 1. Introduction

The African swine fever virus (ASFV) is the pathogen responsible for African swine fever (ASF), a fatal and febrile disease affecting both domestic pigs and wild boars in Eurasia. ASFV is classified within the nucleocytoplasmic large DNA virus group and is the sole representative of the genus *Asfivirus* in the family Asfarviridae [1]. This virus is characterized by its virions, which have a complex, multilayered structure, including the nucleoid at the core, followed by the core shell, inner envelope, capsid and outer envelope, and measure between 175 and 215 nm in diameter [2]. The ASFV genome consists of approximately 190 kbp of double-stranded DNA, containing between 150 and 167 genes, depending on the strain [3]. Historically, ASFVs have been categorized into 23 genotypes based on partial sequences of the B646L gene [4,5]. Recently, a new nomenclature was proposed based on full-length p72 sequences, classifying all known ASFV strains into six distinct groups [6].

ASF poses a global threat to the swine industry due to its transboundary spread. ASF outbreaks over the last decade have caused significant economic losses to pig farmers across the globe. In 2007, a highly virulent genotype II ASFV strain emerged in Georgia and subsequently spread to the Russian Federation, Eastern Europe, China and other Asian countries [7,8,9,10,11,12]. Recently, China also reported the detection of low virulent ASFV strains belonging to p72 genotypes I, complicating the ongoing ASF control measures [13,14].

In response to the demand for an effective ASF vaccine, researchers have developed many live attenuated ASFV strains through genetic manipulation and also by isolating naturally attenuated ASFV from the field [15,16,17,18,19,20,21]. Presently, two such attenuated strains have been licenced and begun to be used to control the spread of ASF in a selected number of countries [18,20]. One of the main concerns of using live attenuated ASFV strains as vaccines is the possibility of the vaccine viruses to recombine between other circulating field strains *in vivo*, giving rise to novel viruses with increased or altered virulence that can evade the protective immunity induced by the vaccines and/or change the clinical signs complicating ASF diagnosis and control.

Recently, China reported detection of chimeric ASFVs (chASFVs) derived from recombination between attenuated ASFV genotype I and circulating virulent genotype II ASFV strains. The chASFVs were resistant to the genotype II based attenuated ASFV vaccine strains [22,23]. Similar chASFV strains have also been reported from Vietnam and Russia, indicating the spread of the emerging chASFV strains into neighbouring countries [24,25].

The chASFVs could be an emerging threat that will impede the control of the ongoing ASF global epidemic. The origin of the emerging chASFVs is not clear, and the mechanisms of genetic recombination between distinct ASFVs remain poorly understood. In this study, we attempted to generate chASFVs both *in vitro* and *in vivo*, using two different attenuated ASFV strains with distinct genotypes, and analyzed their genetic and biological characteristics.

## 2. Materials and Methods

### 2.1. Cells

The IPKM cell line, which is highly permissible to ASFV, was previously established [26,27]. The cells were routinely maintained in Dulbecco modified Eagle medium (Nacalai Tesque, Kyoto, Japan) supplemented with 10% fetal bovine serum, 10 μg/mL bovine insulin (Merck, Darmstadt, Germany), 25 μM monothioglycerol (Wako, Osaka, Japan) and antibiotics in cell culture plates and flasks for suspension culture (Sumitomo Bakelite, Tokyo, Japan).

### 2.2. Viruses

The ASFV OUR T88/3 (genotype I) is a naturally attenuated strain that was kindly provided by Dr. Carrie Batten at the Pirbright Institute, UK [28]. This strain lacks several *MGF* genes as well as *EP153R* and *EP402R* genes, and it does not possess hemadsorption (HAD) activity [29]. The virulent ASFV strain AQS-C-1-22 (genotype II) was isolated from a contaminated pork product confiscated at one of the international airports in Japan [30]. ASFV AQSΔB119L is an HAD positive recombinant virus derived from ASFV AQS-C-1-22, in which the *B119L* (*9GL*) gene was replaced with the green fluorescent marker gene (copGFP). ASFV virus propagation and titrations in IPKM cells and subsequent animal experiments were performed in the biosafety level 3 facility of the National Institute of Animal Health, Tokyo, Japan, which was accredited by the national authority.

### 2.3. Virus Titration

Viral titers were determined using IPKM cells. Briefly, IPKM cells (2 × 10^4^ cells/well) were seeded into 96-well cell culture plates one day before the assay, and one hundred microliters of tenfold serially diluted samples containing ASFV were inoculated into the wells in quadruplicates and incubated for 7 days at 37 °C in a 5% CO_2_ (95% air) incubator. The presence of cytopathic effects (CPEs) was examined under a light microscope, and median tissue culture infectious dose per mL of each sample (TCID_50_/mL) was calculated using the Reed and Muench method [31].

### 2.4. Generation of AQSΔB119L

The GFP expressing B119L gene-deletion mutant of AQS-C-1-22 (AQSΔB119L) was generated by plasmid-based homologous recombination in IPKM cells as described previously [32]. Briefly, the upstream region of the *B119L* (*9GL*) gene, the p72 gene promoter sequence, and the *copGFP* gene, as well as the downstream region of the *B119L* gene, were cloned in this order into the pGEM-T Easy Vector resulting in a plasmid (pTEΔB119L). The plasmid contained *B119L* gene with a 149 nucleotide deletion (amino acid residues 8 to 57) and insertion of the copGFP gene under the p72 gene promoter (Appendix A). Following transfection of pTEΔB119L into IPKM cells, the cells were infected with ASFV AQS-C-1-22 at an MOI of 0.1. The culture supernatant was collected at 5 dpi and subjected to limiting dilution three times and a GFP expressing virus AQSΔB119L was isolated. The purity of AQSΔB119L was confirmed by conventional PCR using the forward primer 5′-AAATCAACATTAACGGCAGC-3′ and reverse primer 5′-AACTTTATCGAGTCTCTGCC-3′ to amplify the *B119L* gene and the same forward primer and an additional reverse primer (5′-TCAGGCGAAGGCGATGGGGGTC-3′) to amplify the *copGFP* gene.

### 2.5. Next-Generation Sequencing of AQSΔB119L

The cell culture supernatant (37 mL) containing AQSΔB119L was centrifuged at 180,000× *g* at 4 °C for 3 h. The resulting pellet was then resuspended in 100 µL of phosphate-buffered saline (PBS), benzonase nuclease (250 U, Merck) was added and the mixture was incubated at 37 °C for 1 h. Viral DNA extraction was performed using a High Pure Viral Nucleic Acid Kit (Roche, Basel, Switzerland) according to the manufacturer’s protocols. The extracted DNA was subjected to whole genome sequencing analysis using the iSEQ 100 platform (Illumina, San Diego, CA, USA). The reads were mapped to AQS-C-1-22 (GenBank accession no. LC659087) using Bowtie2 on the Galaxy web platform v2.3.0 [33].

### 2.6. Growth Kinetics

IPKM cells seeded into 24 well-plates were inoculated with ASFV at a MOI of 0.01. After incubation for 1 h at 37 °C, the inoculum was removed, the cells were washed once and fresh growth medium was added. The culture supernatants were collected every 2 days up to 6 days post inoculation (dpi). TCID_50_ of all collected supernatants was determined using IPKM cells.

### 2.7. In Vitro Generation of chASFVs

IPKM cells grown in T-25 flasks were simultaneously infected with two strains of ASFV, OUR T88/3 (attenuated genotype I) and AQSΔB119L (attenuated genotype II ) at a MOI of 1.0 each. The culture supernatant was collected at 4 dpi, when most of the infected cells showed CPE. Then, the supernatant was serially diluted with the growth medium and transferred to 96-well plates containing IPKM cells and porcine red blood cells. Five days later, the wells that exhibited HAD but no fluorescence were identified; culture supernatant was harvested and subjected to two additional rounds of limiting dilutions, to obtain three isolates named, chASFV1, 2 and 3. The isolates were propagated in IPKM cells, and they continued to show HAD but no green fluorescence. Total nucleic acid extracted from IPKM cells infected with purified chASFV1, 2 and 3 isolates were subjected to whole genome sequencing as described in Section 2.5. The obtained sequence reads were mapped to AQS-C-1-22 (GenBank accession no. LC659087) or OUR T88/3 (GenBank accession no. NC_044957) using Bowtie2 on the Galaxy web platform v2.3.0 [33]. Conventional PCR followed by Sanger sequencing was used to confirm the sequences in the areas with poor coverage.

### 2.8. Animal Experiments

In this study, crossbreed Landrace × Large White × Duroc (LWD) female and castrated male suckling pigs were used. The suckling piglets were used due to the limited biosafety level 3 large animal cubicle space at the National Institute of Animal Health (NIAH), Japan. All the animal experiments followed the guidelines and regulations outlined in the Guide for the Care and Use of Laboratory Animals by the NIAH, National Agriculture and Food Research Organization (NARO), the Guidelines for Proper Conduct of Animal Experiments by the Science Council of Japan [34] and the ARRIVE guidelines [35]. The experimental protocol was reviewed and approved by the Institutional Animal Care and Use Committee at the NIAH, NARO (approval number R4-I020-NIAH, R5-I010-NIAH-2). Throughout the study, the pigs were provided with ample amount of commercial sow milk replacer and water, and efforts were made to ensure their welfare and minimize stress. If a pig exhibited a significant reduction in activity and became recumbent, euthanasia was performed as it reached the humane endpoint.

#### 2.8.1. Evaluation of Virulence of the Parental Viruses and Their Progeny ASFVs Generated *In Vitro*

A total of twelve, 3-day-old suckling pigs were used to assess the virulence of strains OUR T88/3, AQSΔB119L, chASFVs1-3 and AQS-C-1-22. The pigs were obtained from a conventional herd with high health-status and were randomly assigned into six groups (*n* = 2 each). After 7 days of acclimatization period, pigs in each group were inoculated intramuscularly with 10^2^ TCID_50_ of each viral strain per pig.

The pigs were monitored daily for clinical signs and changes in body temperature for up to 6 dpi. The spleen and gastro-hepatic lymph nodes (LNs) were collected from deceased or euthanized pigs at necropsy to examine the copy number of viral genes using an ASFV-specific direct real-time PCR kit.

#### 2.8.2. *In Vivo* Generation of chASFVs

A total of thirteen, 3-day-old suckling pigs were used in this experiment. After 7 days of acclimatization period, four pigs were co-infected with a mixture of OUR T88/3 and AQSΔB119L strains simultaneously into the semitendinosus muscle of the thigh. Control groups were inoculated with low virulent OUR T88/3 (*n* = 3), AQSΔB119L (*n* = 3) or parental virulent AQS-C-1-22 (*n* = 3). Each group was housed in individual pens. All viruses were administered at a dose of 10^2^ TCID_50_ per pig. The pigs were monitored daily for clinical signs and changes in body temperatures. The spleen and gastro-hepatic LNs were collected from deceased or euthanized pigs at necropsy to measure the copy number of viral genes using an ASFV-specific real-time PCR as described below (Section 2.10).

### 2.9. Isolation and Genome Characterization of In Vivo Generated chASFVs

Spleen homogenate was prepared from each pig, and the virus titer of each homogenate was determined using IPKM cells. Following titration, each homogenate was subjected to limiting dilution assay. This procedure was carried out three times, and the isolated viruses were named according to the inoculation groups. For example, the viruses isolated from pig#1 of co-infected group were labeled as coinfected#1 clone 1, 2 and 3. Whole genomes of all isolated viruses were determined in the same way as described in Section 2.7.

### 2.10. Direct Quantitative Real-Time PCR

Spleen and gastro-hepatic LNs collected from dead or euthanized pigs were homogenized in PBS with a Micro Smash homogenizer (TOMY, Tokyo, Japan), and the supernatants were mixed in an equal volume of Lysis Buffer S Ver.2 (TAKARA Bio, Shiga, Japan). After incubation at room temperature for 5 min, the supernatants were directly (without nucleic acid extraction) used in ASFV-specific direct real-time PCR kit (TAKARA Bio, Shiga, Japan) according to the protocol described in our previous study [36].

### 2.11. Phylogenetic Analysis

A neighbor-joining tree based on p72 (*B646L*) gene was generated with the Kimura’s two-parameter model [37]. The phylogenetic tree was generated using MAFFT and MEGA version 7.0 software [38,39]. The accession numbers of sequences used in this tree are described in Appendix A.

### 2.12. Statistics

The data with viral growth kinetics of chASFVs were analyzed using Dunnett’s test to determine the statistical significance of differences.

## 3. Results

### 3.1. Generation of AQSΔB119L

Previous studies have shown that ASFV (ASFV-G-Δ9GL) lacking *B119L* (*9GL*) gene is attenuated both *in vitro* and *in vivo* [32]. In this study, we generated AQSΔB119L by deleting 149 nucleotides of the *B119L* gene and inserting the fluorescent marker protein gene (*copGFP*) into the virulent field strain AQS-C-1-22. The identity of the AQSΔB119L was confirmed by conventional PCR. No amplification was observed from AQSΔB119L with *B119L* specific primers, whereas the amplicon of an expected size (1940 bp) was generated with the *copGFP* specific primers (Figure 1a). In contrast, AQS-C-1-22 genomic DNA resulted in an amplicon of expected size (1136 bp) with *B119L* specific primers but no amplicon with *copGFP* specific primers. The whole genome analysis further confirmed the expected deletion (amino acid residues 8 to 57) in the *B119L* gene and the insertion of *copGFP* gene (Figure 1b). In line with previous findings with ASFV-G-Δ9GL, AQSΔB119L showed attenuated growth kinetics in IPKM cells, resulting in a virus titer 1000 times lower than the parent strain at 4 dpi (Figure 2d).

### 3.2. In Vitro Generation of chASFVs

To examine the potential emergence of genetically recombined viruses under *in vitro* culture conditions, we conducted a co-infection experiment using OUR T88/3 and AQSΔB119L in IPKM cells (Figure 2a). The culture supernatant from IPKM cells co-inoculated with OUR T88/3 and AQSΔB119L was collected at 4 dpi, serially diluted and used to infect fresh IPKM cells in 96-well plates. Most wells showed either no HAD or green fluorescence, or both HAD and green fluorescence. However, a few wells exhibited HAD without any green fluorescent signal. The culture supernatants from these wells after two additional rounds of limiting dilutions resulted in three isolates, chASFV1, 2 and 3, that showed HAD without green fluorescence (Figure 2b).

### 3.3. Genome Characterization of the chASFV1, chASFV2, and chASFV3 Isolates

The chASFV1, 2 and 3 expanded in IPKM cells were subjected to whole genome sequencing as described above for the AQSΔB119L, and the sequences were submitted to the Genbank (LC862888-LC862890). The average sequence coverage obtained for each genome is summarized in Appendix A. The genome of the chASFV3 isolate exhibited the highest genetic similarity to AQSΔB119L. The majority of the chASFV3 genome consists of genes originating from AQSΔB119L, with several short genetic elements from OUR T88/3 inserted at multiple locations throughout the genome (Figure 2c). Surprisingly, chASFV3 had a breakpoint in *p72* gene, making genotyping impossible without defining a new clade (Appendix A). The chASFV1 and chASFV2 isolates showed similar genome rearrangement patterns, in which the 5′ end of the genome was derived from AQSΔB119L and the 3′ end of the genome from OUR T88/3, except the terminal regions. The details of the genomic recombination are summarized in Table 1. These results indicate that genotypically different ASFVs can infect and replicate within the same cell and rearrange their genomes through intergenotypic gene complementation during replication, giving rise to chimeric progeny.

### 3.4. In Vitro Growth Kinetics of chASFVs

Next, we examined the replication efficiencies of the three chASFVs in IPKM cell cultures (Figure 2d). Attenuated strains and virulent AQS-C-1-22 were used as controls in this experiment. The virulent AQS-C-1-22 exhibited the highest rate of replication *in vitro*, and the avirulent OUR T88/3 exhibited the second highest rate of replication. In contrast, the attenuated deletion mutant AQSΔB119L replicated 500-fold less effectively in line with ASFV Georgia 2007 9GL (B119L) deletion mutants [23]. Notably, the replication rates of all three chASFVs were comparable to that of OURT88/3 and 10- to 100-fold higher than AQSΔB119L with significant difference. These data demonstrated that the *in vitro* growth kinetics of the chASFVs were at least partly restored due to genomic chimerization.

### 3.5. Virulence of chASFVs in Suckling Pigs

To assess the virulence of the three *in vitro* generated chimeric viruses, suckling pigs were intramuscularly inoculated with chASFVs1-3, OUR T88/3, AQSΔB119L and AQS-C-1-22 (two pigs per virus at a dose of 10^2^ TCID_50_ per pig). None of the inoculated pigs developed fever. Both pigs inoculated with AQS-C-1-22 developed anorexia and hemorrhagic diarrhea and were found dead on 4 dpi. All pigs inoculated with chASFV developed anorexia and hemorrhagic diarrhea and died or were euthanized by 6 dpi (Figure 3a). One of the two pigs that received chASFV2 died on 5 dpi. On 6 dpi, one of the two pigs that received chASFV1 and both pigs that received chASFV3 died. On 6 dpi, the remaining pigs that received chASFV1 and 2 were euthanized, as they reached the humane end point. In contrast, pigs inoculated with OUR T88/3 or AQSΔB119L showed no clinical signs of ASF and survived until the end of the experimental period (6 dpi).

At necropsy, pigs inoculated with AQS-C-1-22 also showed hemorrhagic lesions in the gastro-hepatic, mandibular and inguinal LNs, but no other lesions were observed. The OUR T88/3- and AQSΔB119L-inoculated pigs showed no pathological lesions. The chASFV-inoculated pigs showed splenomegaly, hemorrhagic LNs, pulmonary edema, interstitial hemorrhages and hemorrhages in the kidneys (Figure 3c and Table 2). The number of viral gene copies in the spleen and gastro-hepatic LNs was 10- to 100-fold higher in the pigs inoculated with chASFVs and AQS-C-1-22 than in those inoculated with the attenuated viruses (Table 2 and Figure 3b). These observations indicated that the phenotype of the attenuated parental viruses was changed to increased virulence in pigs as their genomes recombined to generate chASFVs.

### 3.6. Co-Infection Experiment in Pigs

Pigs inoculated with OUR T88/3 or AQSΔB119L developed no clinical signs and survived until the end of the experiment (6 dpi). In contrast, all three pigs inoculated with the parental virulent AQS-C-1-22 strain developed anorexia and died within 5 dpi (Figure 4b). No clear pyrexia was observed in those pigs. The four pigs that were intramuscularly inoculated with both OUR T88/3 and AQSΔB119L viruses also developed clinical signs, but their onset was delayed. Pigs#1, #2 and #4 exhibited pyrexia, anorexia and diarrhea on 6 dpi and were euthanized on 9 dpi as they reached the humane end point. Pig#3 was found dead on 6 dpi without developing any clinical signs.

During post-mortem inspection, pigs inoculated with either OUR T88/3 or AQSΔB119L showed no pathological lesions (Figure 4c). Conversely, all pigs in the co-infected group exhibited splenomegaly, pulmonary edema and severe hemorrhagic lesions in the submandibular and abdominal LNs. Pigs inoculated with AQS-C-1-22 showed hemorrhagic lesions in the internal organs and the abdominal LNs without splenomegaly. The enhanced disease manifestations in pigs co-infected with attenuated strains suggest possible *in vivo* recombination resulting in viruses with enhanced virulence within days post infection.

### 3.7. Isolation and Genome Characterization of Chimeric ASFVs Generated In Vivo

In order to isolate possible chASFVs from pigs inoculated with OUR T88/3 and AQSΔB119L, spleen homogenates from four co-infected pigs were subjected to repetitive limiting dilution assays. After three limiting dilution assays, three independent chASFVs were obtained from each spleen homogenate, and subjected to whole genome sequencing. Their whole genome sequence data were deposited in the GenBank under the accession numbers LC862876-LC862887. The average coverage was described in Appendix A. The whole genome sequences revealed striking construct between the genomes (Figure 5a). In contrast, the whole genome sequence data of ASFVs isolated from pigs that were inoculated with single attenuated viruses were exactly identical to those of the original viruses.

The genome sequences of twelve chASFVs generated *in vivo* displayed a marked diversity with various chimeric patterns. The genetic rearrangements were most frequently identified at the flanking regions of ORFs related to virus attenuation (Table 3). Such mosaic patterns were confirmed by Sanger sequencing analysis (Figure 5b). These findings highlight the emergence of novel chimeric virulent ASFVs with diverse genome sequences, generated through genetic rearrangement between simultaneously inoculated, genetically distinct attenuated ASFV viruses *in vivo*.

## 4. Discussion

Since 2007, global efforts have been made to develop a safe and effective vaccine against genotype II ASFV that can be used to control outbreaks in Europe, Russia and Asia. Meanwhile, simultaneous occurrence of genotype I ASFV was reported in China, leading to the later emergence of highly virulent chimeric genotype I/II viruses. These viruses have shown resistance to genotype II-based vaccines, posing new challenges for disease control [13,22]. The chimeric genotype I/II viruses have now spread to Vietnam and Russia [24]. As a result, implementing multivalent strategies to control both genotype II and chimeric viruses seems to be crucial. The origin of these chimeric viruses is unclear; however, potential co-infection of pigs already infected with low virulent OURT/88/3 like virus with ASFV genotype II low virulent or highly virulent strain is hypothesized. In order to test this hypothesis, we experimentally co-infected an ASFV permissive cell line and pigs with two genetically different attenuated ASFV strains, ASFV OURT/88/3 and AQSΔB119L.

AQSΔB119L was generated from deletion of the *B119L* (*9GL*) gene from the highly virulent AQS-C-1-22 (genotype II) strain, as described previously with slight modification [32]. Since the *B119L* gene is located immediately upstream of the *B962L* gene, we only deleted the upstream portion of the *B119L* gene (50 amino acids), between positions 8 and 57, to avoid disrupting the promoter of the downstream gene. As demonstrated in the present report, AQSΔB119L replicated less effectively in IPKM cells (Figure 2d) and exhibited decreased virulence in pigs (Figure 3 and Figure 4). These results suggest that the deletion in the *B119L* gene was sufficient to attenuate the highly virulent ASFV AQS-C-1-22 (genotype II) strain. These findings are in line with those observed with ASFV-G-Δ9GL [32]. ASFV-G-Δ9GL replicated poorly in primary swine macrophage cell cultures, and when inoculated intramuscularly into pigs at 10^2^ or 10^3^ HAD_50_ per pig, they did not show any signs of clinical disease for 21 days. 

In this study, we demonstrated that chimeric viruses with altered virulence can be generated through co-infection of genotype-unmatched ASFVs, both *in vitro* and *in vivo*. The isolation of chimeric viruses generated in the cell culture was conducted by visual screening based on their HAD-positive, GFP-negative phenotype, which indicates genetic exchange between the parental viruses. This screening approach likely led to the biased selection of mutant viruses with condensed breakpoints flanking the upstream and downstream of *CD2v* and *B119L* genes. It is possible that many other chASFVs were generated in IPKM cells co-infected with the attenuated ASFVs, but they were not identified in this study.

In the animal experiments, chASFVs generated *in vitro* and in co-infected suckling pigs exhibited enhanced pathogenic phenotypes. In these assays, chimeric viruses, which possess a better replication efficiency and higher virulence due to high frequent recombination flanking MGF, *CD2v* and *B119L* genes, are to be identified due to technical limitation for detecting recombinant viruses. Although the exact process of genomic recombination remains under investigation, genetic rearrangements between ASFVs observed in this study seem to occur randomly in size and position. This observation suggests that this method can be applied to generate a variety of randomly recombined virus strains.

In the co-inoculation study, pigs infected with two different attenuated viruses exhibited typical manifestations of ASF and either died or reached the humane endpoint. It is noteworthy that none of suckling pigs, including those that received the highly virulent ASFV AQS-C-1-22, developed fever. This is in contrast to our previous observations in weaned piglets infected with ASFV AQS-C-1-22 [22]. The exact reason for the absence of fever remains unclear; however, it could be due to per-acute disease induced by the highly virulent strains in suckling piglets. The parental OUR T88/3 is defective of several *MGF* and *CD2v* genes, while AQSΔB119L is attenuated due to the partial deletion of *B119L* gene, resulting in reduced proliferation in pigs when inoculated at low doses. Therefore, clinical disease exhibited in pigs received both attenuated strains that strongly suggest the emergence of virulent chASFVs. In fact, genome analysis of three randomly isolated viruses from splenic emulsion of each pig showed that all viral strains studied had undergone genomic recombination between the parental viruses. This implies that these chimeric viruses become a major viral population in affected animals and gained increased virulence which caused severe consequences. As indicated in this study, genomic rearrangements between avirulent parental viruses may occur between proximal regions of respective deleted genes, *CD2v* and *B119L*, at higher frequencies compared with other regions. In other words, by utilizing different attenuated viruses with distinct genetic backgrounds, it is possible to generate chimeric mutants of ASFV with variably recombined genomes.

*In vivo* recombination events observed in this study between OUR T88/3 and AQSΔB119L support the theory that the recombinant genotype I/II strains that emerged in China could have been originated from a pig coinfected with a low-virulent genotype I virus and a low or high virulent genotype II strain. Two different low virulent non-hemadsorbing genotype I ASFV strains that are closely related OUR T88/3 and low virulent genotype II viruses have been detected in swine herds in China [14,40]. The high frequency of *in vitro* and *in vivo* recombination events observed in this study could be due to close phylogenetic similarity between genotype I and II viruses. In sub-Saharan Africa, many different ASFV genotypes circulates, and therefore future studies should focus on possible recombination between the common ASFV genotypes that are co-circulating in the region.

## 5. Conclusions

ASF live attenuated virus strains (LAVs) are presently the most promising ASFV vaccines [41]. However, our findings raise a significant concern for their use in field conditions. Based on our studies, emergence of revertant viruses with increased virulence because of genetic rearrangement due to functional complementation among circulating ASFVs or with vaccine strains could occur rapidly, subsequently altering the pathogenesis and disease transmission. Furthermore, those revertant viruses may evade host immunity induced by LAVs, undermining countermeasures against ASF. Therefore, careful consideration on the pros and cons of field application of LAVs, the implementation of intensive field surveillance and the establishment of a robust and unwavering vigilance system are essentially required. Only through these measures can we confidently and effectively navigate the control of ASF.

## Figures and Tables

**Figure 1 vaccines-13-00462-f001:**
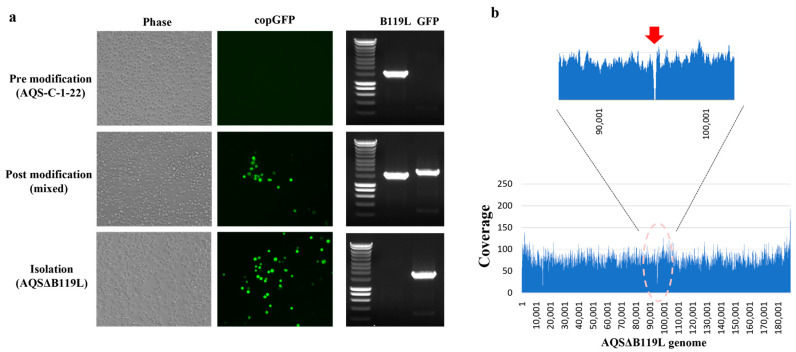
**Generation of an AQSΔB119L strain**. (**a**) IPKM cells inoculated with either the virulent AQS-C-1-22 (pre-modification), supernatant from pTEΔB119L transfected + AQS-C-1-22 infected IPKM supernatant (mixed), or with the purified AQSΔB119L (Isolation). The infected cells were observed at 3 dpi under a light and fluorescent microscope (for copGFP). At 3 dpi, the culture supernatant was harvested, nucleic acid extracted and subjected to conventional PCR using primers specific for *B119L* and *copGFP* genes. (**b**) Next-generation sequencing was performed to determine the whole genome of AQSΔB119L and confirm deletion of *B119L* gene and insertion of *copGFP* gene. The arrow indicates the location of the deletion and the *copGFP* insertion in the whole genome of AQSΔB119L [23].

**Figure 2 vaccines-13-00462-f002:**
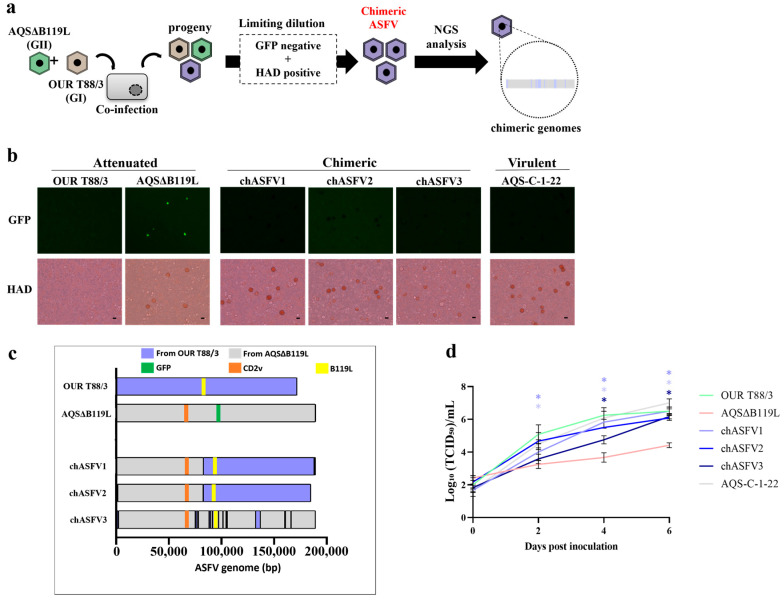
***In vitro* generation, isolation and characterization of chASFVs.** (**a**) Flow diagram of steps followed in *in vitro* generation of chASFVs. IPKM cells were coinfected with OUR T88/3 and AQSΔB119L at a MOI of 1.0, and chimeric viruses showing HAD and not expressing green fluorescence were actively isolated from the culture supernatant by three rounds of limiting dilution and subjected to whole genome analysis. (**b**) Phenotypic characters of OUR T88/3, AQSΔB119L, chASFV1-3 and AQS-C-1-22 when inoculated into IPKM cells. Porcine red blood cells were added immediately after infection to observe the ability of the viruses to induce HAD. Twenty-four hours after inoculation, green fluorescence signals and HAD were observed. Scale bar: 25 µm. (**c**) Next-generation sequencing was performed to determine the genome structure of chASFVs. The sequence reads were mapped to the whole genome sequences of AQSΔB119L and OUR T88/3 or assembled *de novo* to obtain the genome sequences of the respective chASFVs. Orange, green and yellow colors in the map indicate the position of *EP402R*, *copGFP* and *B119L* genes, respectively. The nucleotide sequences originated from AQSΔB119L and OUR T88/3 are shown in gray and blue, respectively. Please note that it was not possible to determine the origin of potential recombination sites with identical nucleotide sequences between OUR T88/3 and AQSΔB119L. Those regions were assigned the same genotype of the sequence prior to the identical region. (**d**) IPKM cells were inoculated with OUR T88/3, AQSΔB119L, the three chASFVs, and AQS-C-1-22 at an MOI of 0.01. The culture supernatants were collected every 2 days and titrated on IPKM until 6 dpi. Data are reported as the mean values ± standard deviation from three independent experiments. Asterisks (*) reveal significant differences compared to AQSΔB119L (*p* < 0.05 by *Dunnett*’s test).

**Figure 3 vaccines-13-00462-f003:**
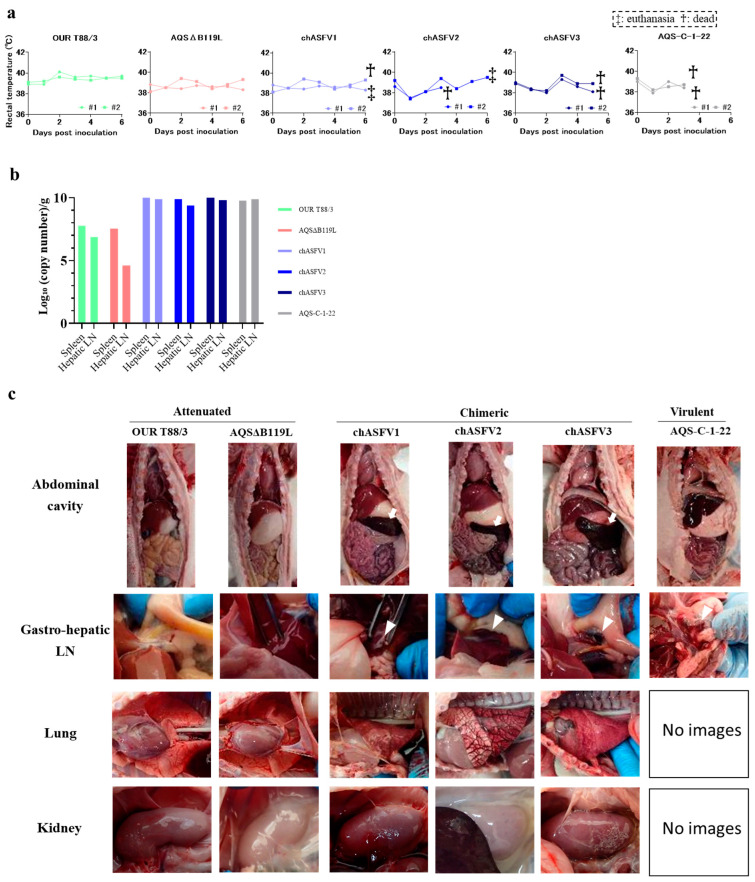
***In vivo* characterization of *in vitro*-derived chASFVs**. (**a**) Rectal temperatures of pigs inoculated with OUR T88/3, AQSΔB119L, chASFVs or AQS-C-1-22 were checked daily until 6 dpi. A temperature above 40 °C for two consecutive days was defined as pyrexia. Crosses and double crosses indicate found dead and euthanatized pigs, respectively. (**b**) Viral gene copy numbers in tissue homogenates of spleen and Gastro-hepatic lymph nodes (LNs) were quantified using an ASFV-specific direct real-time PCR kit (TAKARA Bio. Shiga, Japan). Data are represented as the mean values for each viral strain (*n* = 2). (**c**) Intra-abdominal appearance of pigs infected with each strain of ASFV. Pigs inoculated with attenuated strains showed gross pathological lesions, but those inoculated with chASFVs or the virulent strain AQS-C-1-22 showed hemorrhagic lesions (White arrowhead) in multiple organs (Gastro-hepatic LN, Lung and Kidney). White arrows indicate splenomegaly.

**Figure 4 vaccines-13-00462-f004:**
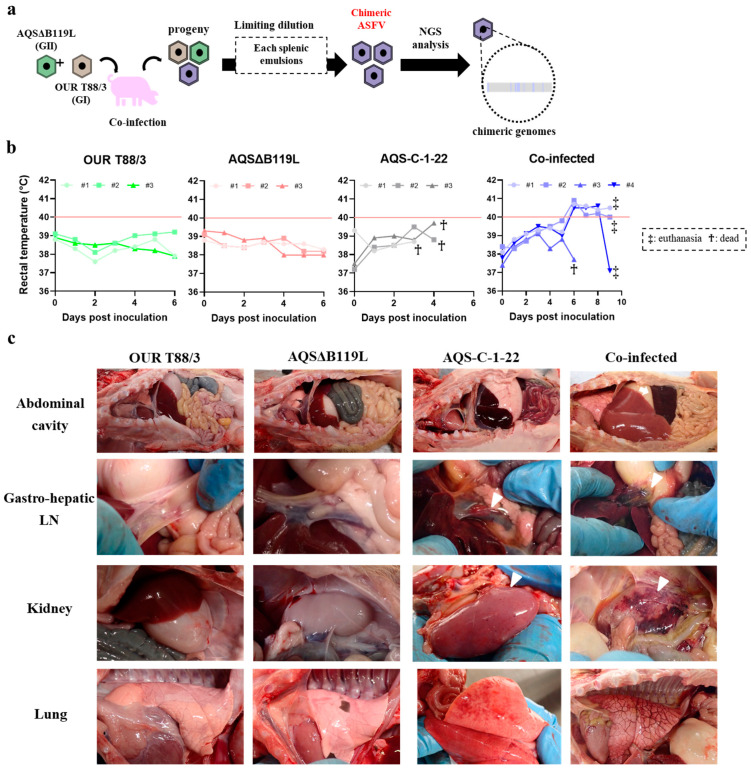
**Experimental co-infection of genetically distinct ASFV strains**. (**a**) Flow diagram describing the *in vivo* generation of chASFVs. Pigs were simultaneously infected with 10^2^ TCID_50_ of OUR T88/3 and 10^2^ TCID_50_ of AQSΔB119L intramuscularly. As control groups, pigs were inocu-lated with either 10^2^ TCID_50_ of OUR T88/3, 10^2^ TCID_50_ of AQSΔB119L or 10^2^ TCID_50_ of AQS-C-1-22. Each spleen sample collected from dead/euthanized pigs was subjected to multiple rounds of limiting dilution to isolate purified virus isolates. Three randomly selected colonies from spleen tissues from each animal were subjected to genome analysis. (**b**) Rectal temperature of pigs inoculated with either OUR T88/3 or AQSΔB119L, and co-infected with OUR T88/3 and AQSΔB119L. Pyrexia was defined by a temperature exceeding 40 °C for two consecutive days. Crosses and double crosses indicate deceased and euthanized pigs, respectively. (**c**) Gross lesions of ASFV-inoculated pigs. Pigs inoculated with a single attenuated strain showed no gross lesions, whereas those inoculated with the virulent strain AQS-C-1-22 or co-infected with both attenuated strains exhibited hemorrhagic lesions (white arrowheads) in multiple organs (gastro-hepatic LN, kidney, lung). A white arrow indicates splenomegaly.

**Figure 5 vaccines-13-00462-f005:**
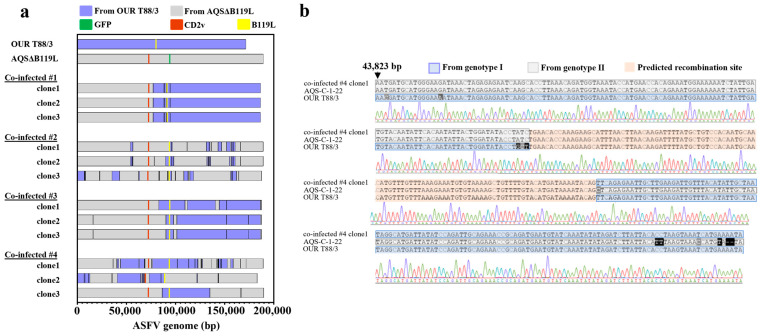
**Genome characterization of chASFVs isolated from pigs co-infected with genetically distinct attenuated ASFV strains**. (**a**) Schematic genome structures of **chASFVs** recovered from four pigs co-infected with both AQSΔB119L and OUR T88/3 strains. Three ASFV clones isolated from spleen homogenate from each pig using limiting dilution method were subjected to whole-genome sequencing. Sequence reads were mapped to the genome sequences of the AQSΔB119L and OUR T88/3 strains. The GFP, CD2v and B119L sequences are shown in green, orange and yellow, respectively. The nucleotide sequences originating from AQSΔB119L and OUR T88/3 are shown in grey and blue, respectively. Please note that it was not possible to determine the origin of potential recombination sites with identical nucleotide sequences between OUR T88/3 and AQSΔB119L. Those regions were assigned to the same genotype of the sequence prior to the identical region. (**b**) A representative recombination site sequence. The genome of co-infected #4 clone1 showed recombination in the region 43,823–44,180 bp. The signal wave remained unmixed throughout this region. The whole genome sequences of the 12 *in vivo* derived chASFVs were submitted to GenBank (LC862876-LC862887). We performed Sanger sequencing of two representative recombinant sites in the genomes of all chASFVs.

**Table 1 vaccines-13-00462-t001:** Genomic information of *in vitro*-derived chASFVs.

Virus	Nucleotide Position	Origin	Genomic Region *
**chASFV1**	1–83122	GII	5′ terminal–C717R
	83123–186605	GI	C717R–NCR
	186606–187759	GII	NCR–3′ terminal
**chASFV2**	1–1197	GI	5′ terminal–MGF_360_1L
	1198–82386	GII	MGF_360_1L–C717R
	82387–186004	GI	C717R–3′ terminal
**chASFV3**	1–1071	GII	5′ terminal–MGF_360_1L
	1072–2285	GI	MGF_360_1L–MGF_360_2L
	2286–2435	GII	MGF_360–2L
	2436–2528	GI	MGF_360–2L
	2529–75440	GII	MGF_360-2L–EP364R
	75441–76884	GI	EP364R–M1249L
	76885–78237	GII	M1249L
	78238–78459	GI	M1249L
	78460–88365	GII	M1249L–C147R
	88366–88562	GI	C147R
	88563–88638	GII	C147R
	88639–89909	GI	NCR–C962R
	89910–91610	GII	C962R
	91611–97563	GI	C962R–B169L
	97564–101679	GII	B169L–B602L
	101680–101746	GI	B602L
	101747–104522	GII	B602L–B646L
	104523–104825	GI	B646L
	104826–105667	GII	B646L–B125R
	105668–105841	GI	B125R
	105842–132741	GII	B125R–NP1450L
	132742–137134	GI	NP1450L–NP868R
	137135–160347	GII	NP868R–QP509L
	160348–161000	GI	QP509L–QP383R
	161001–166080	GII	QP509L–E199L
	166081–166277	GI	E199L
	166278–189355	GII	E199L–3′ terminal

GII: genotype II, GI: genotype I. * Genes on the 5′ and 3′ ends of recombination region or gene within its region.

**Table 2 vaccines-13-00462-t002:** Postmortem lesions observed in pigs inoculated with ASFV highly virulent AQS-C-1-22, attenuated OUR T88/3 and AQSΔB119L, and three *in vitro* generated chASFVs.

Virus	Splenomegaly	Hemorrhagic Lymphadenopathy			Hemorrhagic Tonsils	Pulmonary Edema	Intestinal Hemorrhage
		Gastro-Hepatic L.N.	Mandibular L.N.	Inguinal L.N.			
OUR T88/3	0/2 *	0/2	0/2	0/2	0/2	0/2	0/2
AQSΔB119L	0/2	0/2	0/2	0/2	0/2	0/2	0/2
chASFV1	2/2	2/2	2/2	2/2	2/2	1/2	2/2
chASFV2	2/2	2/2	2/2	2/2	2/2	2/2	2/2
chASFV3	2/2	2/2	2/2	2/2	2/2	2/2	2/2
AQS-C-1-22	0/2	2/2	2/2	2/2	2/2	2/2	2/2

* Table presents number of pigs with lesion/number of sampled pigs.

**Table 3 vaccines-13-00462-t003:** Genomic information of *in vivo*-derived chASFVs.

Virus	Nucleotide Position	Origin	Genomic Region *
**Co-infected #1** **clone1, 2 and 3** **(p72 genotype: I)**	1–77214	GII	5′ terminal–M1249L
	77215–88396	GI	M1249L–C147L
	88397–88541	GII	C147L
	88542–90787	GI	C147L–C962R
	90788–90899	GII	C962R
	90900–94547	GI	C962R–B962L
	94548–94586	GII	B962L
	94587–186639	GI	B962L–3′ terminal
**Co-infected #2** **clone1** **(p72 genotype: II)**	1–54138	GII	5′ terminal–A179L
	54139–56662	GI	A179L–F334L
	56663–93571	GII	F334L–B962L
	93572–95883	GI	B962L–B318L
	95884–95928	GII	B318L
	95929–96771	GI	B318L–B438L
	96772–111757	GII	B438L–G1340L
	111758–111958	GI	G1340L
	111959–131527	GII	G1340L–NP1450L
	131528–134296	GI	NP1450L–NP419L
	134297–134486	GII	NP419L–NP868R
	134487–135213	GI	NP868R
	135214–136589	GII	NP868R
	136590–136770	GI	NP868R
	136771–143426	GII	NP868R–D1133L
	143427–146985	GI	D1133L–S273R
	146986–152088	GII	S273R–H359L
	152089–155787	GI	H359L–H240R
	155789–156944	GII	H240R–R298L
	156945–158595	GI	R298L–Q706L
	158596–158738	GII	Q706L
	158739–160796	GI	Q706L–QP383R
	160797–166098	GII	QP383R–E199L
	166099–166295	GI	E199L
	166296–189405	GII	E199L–3′ terminal
**Co-infected #2** **clone2** **(p72 genotype: II)**	1–54136	GII	5′ terminal–A179L
	54137–55673	GI	A179L–F317L
	55674–55840	GII	F317L
	55841–56660	GI	F317L–F334L
	56661–78267	GII	F334L–M1249L
	78268–78489	GI	M1249L
	78490–89835	GII	M1249L–C962R
	89836–95761	GI	C962R–B318L
	95762–95881	GII	B318L
	95882–97645	GI	B318L–B169L
	97646–97758	GII	B169L
	97759–98505	GI	B169L–B475L
	98506–131841	GII	B475L–NP1450L
	131842–132555	GI	NP1450L
	132556–133389	GII	NP1450L
	133390–134041	GI	NP1450L–NP419L
	134042–134665	GII	NP419L
	134666–135764	GI	NP419L–NP868R
	135765–154881	GII	NP868R–H233R
	154882–155541	GI	H233R–H240R
	155542–156944	GII	H240R–R298L
	156945–158595	GI	R298L–Q706L
	158596–158738	GII	Q706L
	158739–160796	GI	Q706L–QP383R
	160797–166098	GII	QP383R–E199L
	166099–166295	GI	E199L
	166296–189405	GII	E199L–3′ terminal
**Co-infected #2** **clone3** **(p72 genotype: II)**	1–424	GII	5′ terminal–NCR
	425–7338	GI	NCR–MGF_110-3L
	7339–7444	GII	MGF_110-3L
	7445–8023	GI	MGF_110-3L–MGF_110-5L
	8024–8100	GII	MGF_110-5L
	8101–8216	GI	MGF_110-5L–NCR
	8217–22677	GII	NCR–MGF_360-9L
	22678–23055	GI	MGF_360-9L
	23056–35346	GII	MGF_360-9L–MGF_505-4R
	35347–43095	GI	MGF_505-4R–MGF_505-10R
	43096–62450	GII	MGF_505-10R–K196R
	62451–63395	GI	K196R–K145R
	63396–83097	GII	K145R–NCR
	83098–83659	GI	NCR–C257L
	83660–92024	GII	C257L–B962L
	92025–92092	GI	B962L
	92093–93007	GII	B962L
	93008–95940	GI	B962L–B169L
	95941–99657	GII	B169L–B602L
	99658–100003	GI	B602L
	100004–100007	GII	B602L
	100008–100083	GI	B602L
	100084–108529	GII	B602L–G1340L
	108530–112873	GI	G1340L–G1211R
	112874–112945	GII	G1211R
	112946–112990	GI	G1211R
	112991–114690	GII	G1211R
	114691–118429	GI	G1211R–CP2475L
	118430–156441	GII	CP2475L–Q706L
	156442–157553	GI	Q706L
	157554–158007	GII	Q706L–QP509L
	158008–162049	GI	QP509L–E423R
	162050–164412	GII	E423R–E199L
	164413–164609	GI	E199L
	164610–187722	GII	E199L–3′ terminal
**Co-infected #3** **clone1** **(p72 genotype: I)**	1–83066	GII	5′ terminal–C717R
	83067–109680	GI	C717R–G1340L
	109681–112182	GII	G1340L
	112183–140703	GI	G1340L–NCR
	140704–144505	GII	NCR–D117L
	144506–151970	GI	D117L–H359L
	151971–152084	GII	H359L
	152085–186548	GI	H359L–NCR
	186549–187703	GII	NCR–3′ terminal
**Co-infected #3** **clone2** **(p72 genotype: I)**	1–15933	GII	5′ terminal–MGF_360-4L
	15934–16137	GI	MGF_360-4L
	16138–90233	GII	MGF_360-4L–C962R
	90234–98022	GI	C962R–B169L
	98023–101389	GII	B169L–B602L
	101390–152011	GI	B602L–H359L
	152012–152089	GII	H359L
	152090–174200	GI	H359L–NCR
	174201–174273	GII	NCR–I215L
	174274–186590	GI	I215L–MGF_360-19L
	186591–187738	GII	MGF_360-19L–3′ terminal
**Co-infected #3** **clone3** **(p72 genotype: I)**	1–15933	GII	5′ terminal–MGF_360-4L
	15934–16137	GI	MGF_360-4L
	16138–90228	GII	MGF_360-4L–C962R
	90229–98017	GI	C962R–B169L
	98018–101384	GII	B169L–B602L
	101385–152018	GI	B602L–H359L
	152019–152096	GII	H359L
	152097–174294	GI	H359L–I215L
	174295–174303	GII	I215L
	174304–186596	GI	I215L–NCR
	186597–187739	GII	NCR–3′ terminal
**Co-infected #4** **clone1** **(p72 genotype: I)**	1–36439	GII	5′ terminal–MGF_505-4R
	36440–36485	GI	MGF_505-4R
	36486–40323	GII	MGF_505-4R–MGF_505-6R
	40324–40608	GI	MGF_505-6R
	40609–40624	GII	MGF_505-6R
	40625–42314	GI	MGF_505-6R–MGF_505-7R
	42315–44053	GII	MGF_505-7R–MGF_505-9R
	44054–62670	GI	MGF_505-9R–F1055L
	62671–63089	GII	F1055L–NCR
	63090–68490	GI	NCR–EP1242L
	68491–68586	GII	EP1242L
	68587–68952	GI	EP1242L
	68953–75509	GII	EP1242L–EP364R
	75510–75671	GI	EP364R–NCR
	75672–76620	GII	NCR–M1249L
	76621–101763	GI	M1249L–B602L
	101764–101913	GII	B602L
	101914–113462	GI	B602L–G1211R
	113463–113592	GII	G1211R
	113593–119945	GI	G1211R–CP2475L
	119946–119949	GII	CP2475L
	119950–122327	GI	CP2475L
	122328–122352	GII	CP2475L
	122353–123141	GI	CP2475L
	123142–126042	GII	CP2475L–CP530R
	126043–126567	GI	CP530R
	126568–127129	GII	CP530R–CP80R
	127130–127648	GI	CP80R–CP312R
	127649–136810	GII	CP312R–NP868R
	136811–136846	GI	NP868R
	136847–142113	GII	NP868R–D1133L
	142114–142363	GI	D1133L
	142364–143425	GII	D1133L
	143426–143912	GI	D1133L–NCR
	143913–144288	GII	NCR–D117L
	144289–146283	GI	D117L–S183L
	146284–146756	GII	S183L–S273R
	146756–152760	GI	S273R–H171R
	152761–156283	GII	H171R–R298L
	156284–158737	GI	R298L–Q706L
	158738–166097	GII	Q706L–E199L
	166098–166294	GI	E199L
	166295–171902	GII	E199L–I243L
	171903–171911	GI	I243L
	171912–189400	GII	I243L–3′ terminal
**Co-infected #4** **clone2** **(p72 genotype: I)**	1–6955	GI	5′ terminal–MGF_110-4L
	6966–6979	GII	MGF_110-4L
	6980–6993	GI	MGF_110-4L
	6994–7243	GII	MGF_110-4L–NCR
	7244–7824	GI	NCR–NCR
	7825–7901	GII	NCR–NCR
	7902–11414	GI	NCR–NCR
	11415–11555	GII	NCR–MGF_360-4L
	11556–12896	GI	MGF_360-4L
	12897–35253	GII	MGF_360-4L–MGF_505-5R
	35254–35493	GI	MGF_505-5R–NCR
	35494–40818	GII	NCR–NCR
	40819–64686	GI	NCR–EP1242L
	64687–65706	GII	EP1242L
	65707–66603	GI	EP1242L
	66604–67212	GII	EP1242L–EP424R
	67213–68293	GI	EP424R
	68294–73368	GII	EP424R–M1249L
	73369–85271	GI	M1249L–NCR
	85272–85316	GII	NCR–NCR
	85317–88943	GI	NCR–B962L
	88944–121725	GII	B962L–CP2475L
	121726–122372	GI	CP2475L–CP204L
	122373–143423	GII	CP204L–S183L
	143424–143896	GI	S183L–S273R
	143897–183159	GII	S273R–3′ terminal
**Co-infected #4** **clone3** **(p72 genotype: I)**	1–86447	GII	5′ terminal–C475L
	86448–135136	GI	C475L–NP419L
	135137–166569	GII	NP419L–E199L
	166570–166766	GI	E199L
	166767–189878	GII	E199L–3′ terminal

GII: genotype II, GI: genotype I. * Genes on the 5′ and 3′ ends of recombination region or gene within its region.

## Data Availability

The genome information of our isolates was submitted to the GenBank: LC862876-LC862890.

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
