# Peer review of "Generation of Chimeric African Swine Fever Viruses Through In Vitro and In Vivo Intergenotypic Gene Complementation"

_vaccines, 2025, doi:10.3390/vaccines13050462_

Round 1
Reviewer 1 Report
Comments and Suggestions for Authors
Tomoya et al. described the propensity of ASFV genome for recombination. They showed that genotype I and II chASFV with diverse genomes can be easily generated experimentally both in vitro and in vivo. I think this paper is interesting, as well as for understanding the virus evolution in the field. I have some suggestions as mentioned below:
Concerns
1. How defined genome characterization of chASFVs, especially the length of gene fragment from OUR T88/3 or AQSâ–³B119L, how defined predicted recombination site (the same gene fragment among chASFVs, OUR T88/3 and AQSâ–³B119L), these fragments from OUR T88/3 or AQSâ–³B119L in Figure 2c and 5a,and belonged GI or GII in Table 1 and 3?
2. In Table 1 and 3, the length of some chASFV’s gene fragment from GI or GII only has several or dozen nucleotides. How defined these gene fragments, according SNP site?
3. Next-generation sequencing was used for AQSΔB119L and chASFVs, the authors should added the average coverage of chASFVs. That was very important for their genome data quality and accuracy.
4. Lines 344 and 345, “Such mosaic patterns were confirmed by Sanger sequencing analysis”, How many fragments were amplified and sequenced by Sanger sequencing analysis?
5. Lines 96, 187, and Fig.S2. Change “p72” to “p72”.
6. In b of Figure 2, scale bar should be added.
7. Line 163. Change “10” to “13”.
8. Line 265. Remove “h”.
9. Line 267. Change “was” to “were”.
10. Line 302. Change “chASFV” to “chASFVs”.
11. Line 442. Change “is” to “are”.
12. Why did the pigs infected with virulent ASFVs including chASFV-1, -2,-3, and AQS-C-1-22 have no fever? Almost pigs had the rectal temp less than 40 degree.
Author Response
Please see the attachment.
We sincerely thank the reviewers for taking the time to review our manuscript and providing
constructive feedback to improve the manuscript. We have revised the manuscript accordingly
and below shown are the original comments from reviewers and our corresponding responses.
#Reviewer 1
Tomoya et al. described the propensity of ASFV genome for recombination. They showed
that genotype I and II chASFV with diverse genomes can be easily generated experimentally
both in vitro and in vivo. I think this paper is interesting, as well as for understanding the virus
evolution in the field. I have some suggestions as mentioned below:
We thank the reviewer for the kind feedback.
Concerns
1. How defined genome characterization of chASFVs, especially the length of gene fragment
from OUR T88/3 or AQSâ–³B119L, how defined predicted recombination site (the same gene
fragment among chASFVs, OUR T88/3 and AQSâ–³B119L), these fragments from OUR
T88/3 or AQSâ–³B119L in Figure 2c and 5a, and belonged GI or GII in Table 1 and 3?
We thank the reviewer for this question. As you pointed out, for those regions with identical
nucleotide sequences between OUR T88/3 and AQSΔB119L (predicted recombination
sites), it was not possible to accurately determine the origin of the sequences. Those
sequences were given the same genotype of the sequence prior to the identical region. For
example in Fig. 5b, the virus Coinfected#4-1 shares the same nucleotide sequence as
AQSΔB119L up to the end of the predicted recombination site, so it is classified as GII. After
that, until the nucleotide sequence of Coinfected#4-1 matches that of AQSΔB119L again, it
is classified as GI.
The following statement was added to the Figure 2 and figure 5 figure legends to clarify this
(lines 248-251, 374-377). “Please note that it was not possible to determine the origin of
potential recombination sites with identical nucleotide sequences between OUR T88/3 and
AQSΔB119L. Those regions were assigned the same genotype of the sequence prior the
identical region”.
2. In Table 1 and 3, the length of some chASFV’s gene fragment from GI or GII only has
several or dozen nucleotides. How defined these gene fragments, according SNP site?
Thank you for your question, and we agree that some gene fragments are very short. However,
like other parts of the genome we obtained high coverage in these areas, and the sequences
match 100% with the genotype assigned.
3. Next-generation sequencing was used for AQSΔB119L and chASFVs, the authors should
added the average coverage of chASFVs. That was very important for their genome data
quality and accuracy.
Thank you for your critical comment. We added a new table, Supplemental Table 1, which
shows the average coverage of all chASFVs. Additionally, we revised the sentences at lines
183, 261-262, 343 and 468 in the revised manuscript.
4. Lines 344 and 345, “Such mosaic patterns were confirmed by Sanger sequencing analysis”.
How many fragments were amplified and sequenced by Sanger sequencing analysis?
Thank you for your comment. Based on the genomic information obtained from NextGeneration Sequencing, we performed Sanger sequencing of two representative recombinant
sites per genome, for all chASFV genomes reported in this study.
5. Lines 96, 187, and Fig.S2. Change “p72” to “p72”.
Revised as suggested (lines 97, 193 and 469, Fig.S2.)
6. In b of Figure 2, scale bar should be added.
Done (Figure 2b and line 242).
7. Line 163. Change “10” to “13”.
Done (line 167).
8. Line 265. Remove “h”.
Done (line 283).
9. Line 267. Change “was” to “were”.
Done (line 285).
10. Line 302. Change “chASFV” to “chASFVs”.
Done (319).
11. Line 442. Change “is” to “are”.
Done (line 463).
12. Why did the pigs infected with virulent ASFVs including chASFV-1, -2, -3, and AQS-C-
1-22 have no fever? Almost pigs had the rectal temp less than 40 degree.
Thank you for your comment. First, we apologize for the mistake in our description. The pigs
used in this study were not weaned piglets, but suckling pigs (lines 151-152, 157, 160, 167,
286-287, 416, 425-430 in the revised manuscript). In our previous paper (reference 22), we
reported that AQS-C-1-22, like other highly virulent ASFV strains, induces fever in 8-weekold piglets. However, in subsequent studies when suckling piglets were infected with AQS-C1-22, no pyrexia (over 41oC) was observed. We believe the reason for this observation is that
AQS-C-1-22 is more virulent in suckling piglets compared to weaned piglets, resulting in a
per-acute disease-with no fever. The exact reason for the high virulence/absence of fever in
suckling pigs remains unclear, and further research is needed to address it. The reason for
using suckling piglets in our studies is due to the limited large animal cubicle space at the
National Institute of Animal Health (NIAH), Japan.
The following section as added to the Animal experiments section under Materials and
methods to clarify this (lines 142-144).
In this study, crossbreed Landrace × Large White × Duroc (LWD) female and castrated
male suckling pigs were used. The suckling piglets were used due to the limited biosafety
level 3 large animal cubicle space at the National Institute of Animal Health (NIAH), Japan.

Reviewer 2 Report
Comments and Suggestions for Authors
The manuscript by Kitamura and colleagues describes the isolation of chimeric African swine fever virus (ASFV) strains both in vitro and in vivo following co-infection with isolates from different genotypes. The findings are very important to the field, as they show that recombination between different genotypes (at least between genotypes I and II) is a relatively frequent phenomenon. This indicates that vaccination strategies using live attenuated vaccines should be approached with extreme care, to avoid the emergence of new highly virulent isolates.
The manuscript merits publication, but the following points should be addressed by the authors:
1) The data from the in vivo experiments seems to indicate that the animals were inoculated with titres higher than those indicated in the Materials and Methods section (10e2 TCID50). Indeed, the pigs infected with the virulent AQS-C-1-22 died within 4 days, showing no increase in body temperatures. This clinical presentation points towards peracute African swine fever (ASF). Under experimental settings, this form of disease is generally associated with higher inoculation doses. Given that the titres were calculated using IPKM cells, it is possible that they were underestimated. Despite IPKM cells being susceptible to ASFV, the titres obtained using these cells may not reflect the virus titres using primary pig macrophages. Therefore, the authors should present data showing that titres obtained using IPKM cells are similar (or not) to those obtained using primary pig macrophages.
2) Figure 2d: Please add statistics
3) Figure 3a (last graph on the right): from the graph it looks like the deaths occurred on day 3, however in the text it says 4 days. Please correct graph.
4) Table 2: Define lesions in the lymph nodes
5) Line 34: Nucleoid instead of nucleoids
6) Line 37: There are currently 23 known genotypes, as the previously annotated genotype XVIII was shown to be a mixed population of genotypes I and VIII (https://doi.org/10.1128/mra.00067-24)
7) Line 216: This paragraph needs an introductory sentence
8) Line 338: Please rephrase
9) Line 409: Some, but not all, MGFs are absent in OUR T88/3
10) Line 417: please give references for the “other reports”
Comments on the Quality of English LanguageN/A
Author Response
Please see the attachment.
We sincerely thank the reviewers for taking the time to review our manuscript and providing
constructive feedback to improve the manuscript. We have revised the manuscript accordingly
and below shown are the original comments from reviewers and our corresponding responses.
#Reviewer 2
The manuscript by Kitamura and colleagues describes the isolation of chimeric African swine
fever virus (ASFV) strains both in vitro and in vivo following co-infection with isolates from
different genotypes. The findings are very important to the field, as they show that
recombination between different genotypes (at least between genotypes I and II) is a relatively
frequent phenomenon. This indicates that vaccination strategies using live attenuated
vaccines should be approached with extreme care, to avoid the emergence of new highly
virulent isolates.
The manuscript merits publication, but the following points should be addressed by the
authors:
We thank the reviewer for the kind support.
1) The data from the in vivo experiments seems to indicate that the animals were
inoculated with titres higher than those indicated in the Materials and Methods section (10e2
TCID50). Indeed, the pigs infected with the virulent AQS-C-1-22 died within 4 days, showing
no increase in body temperatures. This clinical presentation points towards peracute African
swine fever (ASF). Under experimental settings, this form of disease is generally associated
with higher inoculation doses. Given that the titres were calculated using IPKM cells, it is
possible that they were underestimated. Despite IPKM cells being susceptible to ASFV, the
titres obtained using these cells may not reflect the virus titres using primary pig macrophages.
Therefore, the authors should present data showing that titres obtained using IPKM cells are
similar (or not) to those obtained using primary pig macrophages.
Thank you for your comment. First, we apologize for the mistake in our description. The pigs
used in this study were not weaned piglets, but suckling pigs (lines 157, 160, 167, 286-287,
416 in the revised manuscript). In our previous paper (reference 22), we reported that AQSC-1-22 like other highly virulent ASFV strains induces fever in 8-week-old piglets. However,
in subsequent studies when suckling piglets were infected with AQS-C-1-22, no pyrexia (over
41oC) was observed. We believe that AQS-C-1-22 is more virulent in suckling piglets
compared to weaned piglets, potentially resulting in a per-acute disease-with no fever. The
exact reason for the high virulence/absence of fever remains unclear, and further research is
needed.
The reason for using suckling piglets in our studies is due to the limited large animal cubicle
space at the National Institute of Animal Health (NIAH), Japan. To over the issue, we have
established stackable cages to increase the number of pigs that can be used in our animal
facility. We added a comment to the Material and methods, and to the discussion (lines 142-
144, 151-152, 425-430).
In our previous paper (reference 18), we confirmed that the detection limit of the titration
assay using IPKM cells was nearly identical to that of the classical HAD assay performed with
primary alveolar macrophages.
2) Figure 2d: Please add statistics
Your comment is appreciated. We statistically analyzed the data by Dunnett’s test and
modified Fig. 2d accordingly. We also added the sentences in lines 198-200, 254-257, and 284.
3) Figure 3a (last graph on the right): from the graph it looks like the deaths occurred on
day 3, however, in the text, it says 4 days. Please correct graph.
Thank you for the comment. One of the chASFV2-inoculated pigs and all of the AQS-C-1-
22-inoculated pigs died on day 4 before we measured body temperature. That is why we did
not show the body temperatures of those pigs at 4 dpi.
4) Table 2: Define lesions in the lymph nodes
We added missing information about the definition of the lesions in Table 2.
5) Line 34: Nucleoid instead of nucleoids
Revised as suggested (line 34).
6) Line 37: There are currently 23 known genotypes, as the previously annotated genotype
XVIII was shown to be a mixed population of genotypes I and VIII
(https://doi.org/10.1128/mra.00067-24)
Thank you for your suggestion. We have referenced the paper you provided and made the
corresponding corrections in our manuscript (lines 38 and 500-502).
7) Line 216: This paragraph needs an introductory sentence
Thank you for your suggestion. We added an introductory sentence at lines 225-231.
8) Line 338: Please rephrase
Thank you for the suggestion. We reviewed the content and changed the word “contrast” to
“construct” at line 344.
9) Line 409: Some, but not all, MGFs are absent in OUR T88/3
We revised the sentence at line 430.
10) Line 417: please give references for the “other reports”
Thank you for your comments. The statement about the existence of other references was
incorrect. We deleted these words in the revised manuscript (line 438). We sincerely
apologize for the mistake.
